# Revolution and Nation: Johann Gottlieb Fichte's Late Philosophy of Religion

Christoph Asmuth

Lehrstuhl für Philosophie, Augustana Theological University, 91564 Neuendettelsau, Germany;
christoph.asmuth@augustana.de

**Abstract:** Johann Gottlieb Fichte's philosophy of religion combines revolutionary pathos with Christian convictions and transcendental philosophical insights. The result is a bourgeois philosophy of religion that preaches freedom, equality and brotherhood, expects the national upswing of a still-longed-for Germany based on the example of revolutionary France, and praises all this as a continuation of Kant's philosophy.

**Keywords:** French Revolution; Christianity; nationalism; transcendental philosophy; subjectivity; J.G. Fichte; I. Kant

## 1. Religion and Politics as a Problem in Fichte

Immanuel Hermann Fichte, the only child of Johann Gottlieb and Johanna Fichte and himself a highly influential 19th century philosopher, recounted from his childhood memories that his father led a brief daily prayer meeting. The meeting usually included not only the family, but also the household staff. It began with piano accompaniment and the singing of a chorale. After this musical prelude, old Fichte would address the congregation and reflect on a passage from the Bible, especially the Gospel of John. Occasionally, his reflections would be interspersed with words of comfort or exhortation, depending on the occasion (Fichte 1862, p. 428f).

In addition to this testimony of Immanuel Hermann, which likely pertains to Fichte's time in Berlin, there are numerous sources attesting that the elder Fichte was deeply rooted in the evangelical-Lutheran pietistic tradition (Cf. Traub 2020). He himself took on the role of a preacher. His popular speeches on religion and politics, which later at least partially found expression in popular books, are rhetorically indebted to the style of preaching.

The edition of his father's writings, which Immanuel Hermann compiled after the father's death, reinforces the image of a deeply religious philosopher who struggled throughout his life for the Absolute, i.e., God. This struggle is an expression of both the fear of God's essence and the will to understand the world and nature only from a divine perspective. Immanuel Hermann has left traces suggesting that his father can be deciphered as a religious mystic concerned with the contemplative reflection (*Einkehr*) of the divine. This impression is strengthened by the fact that Immanuel Herrmann's edition divides Fichte's writings into special sections: The Theoretical Philosophy appears in its own section, separate from the Popular Philosophy and the Political Writings. Such a division is modern. It corresponds to the disciplining of philosophy, a process of differentiation that already gained momentum in the 19th century. It disciplines—and thus defuses—Fichte's philosophy, confining it to a corset of intra-philosophical disciplinary boundaries.

The editorial tendency of his father's writings is clear: to keep Fichte out of the ideological, political, and religious controversies of his time. Such a tendency attempts to impose a timeless, ahistorical position on Johann Gottlieb Fichte, which—especially as far as Fichte's so-called late philosophy is concerned—bears much resemblance to a pre-critical, rationalist metaphysics of being. In particular, the Atheism Controversy is defused in terms

of its escalation and impact. At the center of Fichte's philosophy is the *Wissenschaftslehre*, the religious significance of which is emphasized, while the popular philosophy seems to be marginalized: the popular treatises are occasional writings whose political significance lies primarily in the nationalist spirit they seek to spread.

Recent research, by contrast, rightly outlines the *whole* Fichte (Cf. Oesterreich and Traub 2013; Zöller 2013). Such an outline has been made possible by the fact that the complete edition of Johann Gottlieb Fichte's writings, which has been available for several years now, follows a strictly chronological approach and—in contrast to the edition selected by Fritz Medicus at the beginning of the 19th century—also includes all existing texts. Following such a synthetic view of Fichte's work, it is no longer surprising that, according to his son's testimony, the private Fichte attached great importance to a lively religiosity during his time in Berlin, although he had been accused of atheism a few years earlier.

This essay attempts to understand Fichte's late philosophy of religion as an integral part of his philosophical work, which was more strongly influenced by the course of time than Kant's work, for example. It is important to bear in mind that Fichte did not want to pursue a book philosophy that progressed in propositions, but rather a practical philosophy of freedom (Cf. Asmuth 2007b; Danz 1997, 2012). His doctrine of science is aimed at action, the transition into life: lived morality. His philosophy of religion therefore developed into a political theology in response to the political events of the French Revolution and the Napoleonic Wars. His work thus fell into a field of tension between his fundamental pietistic convictions, his support for the French Revolution, his rejection of feudalism and monarchism, and his desire to see Germany become a nation.

## 2. The So-Called Atheism Controversy

The starting point of the Atheism Controversy, which from a journalistic point of view was extremely far-reaching, was the religious-philosophical work *On the Ground of Our Faith in a Divine Government of the World* (*Über den Grund unseres Glaubens an eine göttliche Weltregierung*). It must be admitted, however, that Fichte's work had little influence on the confrontation that ensued. A sum of very different interests, moral and religious ideas, political attitudes, and personal conflict strategies determined the direction and severity of the confrontation. Fichte's argument consisted in presenting a philosophical concept of God that conceived of the Absolute as a *moral world-order* (*moralische Weltordnung*). The reduction of God to the realm of morality made Fichte the spearhead of a modern theology that jettisoned the metaphysical foundation and saw religion as a practical dimension of human existence. Such a reduction meant that traditional theological attributes of God were suspended, such as the personality of God, His transcendence, the immortality of the soul, the forgiveness of sins, and so on. Fichte's new understanding of God was even offensive because it transformed Kant's critical reservations—namely, that the existence of God could not be proven, and that the immortality of the soul could not be discussed scientifically—into positive but dismissive assertions. Those who already sensed danger for orthodox Christianity in Kant's critical philosophy read Fichte's reduction of theology to morality as confirmation of their suspicions (Danz 1999).

Fichte's philosophy originally drew on various sources. At first, *before* he adopted Kant's Critical Philosophy, Fichte was inclined towards causal determinism (Cf. Wildfeuer 1999). However, morality and responsibility could not be justified in a deterministic framework, as the freedom of the individual would then be a mere appearance and the will would be reduced to a mere state of consciousness. At the same time, determinism arose from a political feeling of despair: truly *profound* changes to the social situation could not be caused by the individual's ability to act. A consistent causal determinism views society and its progress as analogous to natural entities that are determined by regular natural forces. Fichte himself outlines a philosophical representation of that position in his treatise *The Vocation of Man* (*Bestimmung des Menschen*). In this text, Fichte characterizes determinism as a worldview to be overcome, an overcoming that is only possible through the immediate awareness and feeling of freedom.

Fichte discovered such a consciousness of freedom in Kant's practical philosophy. The evidence we have suggests that Fichte's transformation from determinism to freedom-consciousness took place in August 1790 (Asmuth 2001). For him, the transformation also marked the end of his political concession. Looking at Fichte's development from this synthetic perspective, one can no longer separate his philosophy of religion from his political philosophy. Fichte's early political philosophy is entirely focused on the French Revolution. In a way, it is astonishing that Fichte continues to defend the Revolution for so long, even when it becomes clear that the Revolution is turning into a Reign of Terror, as Georg Büchner puts it in the mouth of his *Danton* on the scaffold: "I know well—the Revolution is like Saturn, it eats its own children." (Büchner 1835) Of course, Fichte does not defend the Reign of Terror, but he does defend the aims and the necessity of a bourgeois revolution. The transformation also changed Fichte's view of the theological situation of his time. *Freedom* is the keyword under which his philosophy can now be subsumed (Cf. Fichte et al. 1973, p. 182). Fichte's contemporaries also deciphered his groundbreaking work, with the programmatic title *Foundations of the Entire Wissenschaftslehre* (*Grundlage der gesammten Wissenschaftslehre*), within this context. Although Fichte did not speak of God in this treatise, the argumentative figure of the self-positing I is revealed to his readers at the same time the empowerment of the subject over God and the worldly ruler. And in the formulation that God is nothing other than the moral world order, they saw the removal of God and His transformation into an inner-worldly morality: an atheism through the secularization of God! Ultimately, Fichte's conception in Jena and later in Berlin is very similar to the religious cults of the French Revolution, the cult of reason (Hebertists) or, for example, the cult of the Supreme Being, which was inaugurated by Robespierre in the spring of 1794, but had already been laid down in the preamble to the Rights of Man and of the Citizen in 1789. The cult of the Supreme Being was intended to break the hegemony of Catholicism in France, but also to prevent atheism while guaranteeing religious freedom. Since his time in Jena, Fichte's philosophy of religion had been fused with political motives. The image of the private man and citizen at the piano, singing a chorale for his family and home, emphasizes the philosopher's personal piety. But Fichte's concern was different! He was concerned with the development of a political theology.

### 3. The Wissenschaftslehre in Berlin

After the Jena system, Fichte underwent two further important phases in the consolidation of his philosophy. The first phase took place in the years 1804/05 (Cf. Asmuth 2007a), when Fichte lectured four times on the *Wissenschaftslehre*, three times in Berlin, and one semester in Erlangen. In this framework, he also lectured on the principles of God, morality and right (*Sitten- und Rechtslehre*), as well as on logic and metaphysics. In his popular lectures in Berlin during these years, Fichte devoted himself to the *Essence of the Scholar* (philosophy of education), the *Characteristics of the Present Age* (philosophy of history), and *The Way Towards The Blessed Life* (doctrine of religion).

Without relying on any particular one of the numerous versions of the *Wissenschaftslehre*, in the following I will summarize the main ideas of the Berlin *Wissenschaftslehre*. First of all, it must be emphasized, especially in view of the frequent use of the word 'Absolute', that Fichte's Berlin *Wissenschaftslehre* focuses on the *finite rational being*, humans, who assure themselves of their infinite ground in the Absolute through rational finite thinking. But in Fichte's view, the Absolute is not a metaphysical, superhuman entity, but nothing other than reality itself, which for Fichte is the rational in its absolute unity. By turning to this rational ground through *their own* reason, human beings, who remain finite in their finiteness, discover their infinity. Such infinity is more than what every finite human being find in himself or herself. Since such infinity is common to all individuals, there is a unity and rationality that unites all and everyone into one humanity, into one reason—a philosophical foundation of "freedom, equality, brotherhood". Reality, unity, and reason stand under a unity focus, and are themselves nothing other than unity—a legacy of Kant's transcendental apperception and the 'I' of the early *Wissenschaftslehre*.

Therefore, all difference is a difference of form, but not of one absolute content. Difference is a matter of view, difference is a matter of perspective; Fichte's philosophy is therefore a theory of perspective referring to itself. Knowledge is always fanned out in perspective, but, insofar as it is knowledge, it is committed to unity. In this, Fichte's concern differs from that of Hegel, whose system construction always includes the overview., i.e., the perspective of the whole.[1] Fichte, on the other hand, insists that even the overview is only a certain view of the whole, but by no means the whole itself. In addition to the unity of *the Absolute*, there is therefore the duplicity between the *view of the Absolute* and the multiplicity *of views of the Absolute*, a moment of tension that characterizes Fichte's entire philosophy. It is in this tension that the duplicity of the ideal and the real and their ideal duplication in idealism and realism belong (Cf. Zöller 2006, 2008; Further: Asmuth 1999, 2009). It also includes Fichte's theory of the fivefold world view, in which the world with its infinite forms and shapes is viewed in a fivefold way, world-views (*Weltansichten*), which underlie all experience a priori.

In doing so, Fichte always remains committed to the program of transcendental philosophy. Regardless of how one views Fichte's philosophical development, whether he *distances* himself from his philosophical beginnings and from Kant's critical philosophy, whether he dissociates himself from the philosophy of the I, which he elaborated in the *Foundations of the entire Wissenschaftslehre*, or whether he fundamentally *revises* the entire Jena system, one thing should be clear: there is no explicit textual evidence that he restructured his transcendental philosophical concept in favor of an ontology or metaphysics, for example. On the contrary, the late *Wissenschaftslehre*, paradigmatically and explicitly the *Wissenschaftslehre of 1810*, radicalizes transcendental philosophy and develops a new variant of this line of argument.

Kant's critique of reason searches for the conditions of possibility of cognition and, as a result, obtains a construct of conditions of possibility without an ontological substrate, the validity of which is independent of both empirical conditions and metaphysical-theological presuppositions. However, whenever there is real cognition, the conditions of possibility are also given, i.e., real (*wirklich*) and—in relation to the knowledge of rules—valid (*gültig*).[2]

Fichte's *Wissenschaftslehre* stands under the primacy of practical reason: Knowledge itself is practical, is a living process. There is no substantial difference between philosophy and revolution. At the same time, the contrast between theory and practice is implemented and dissolved in the interrelationship between the I and the world. The *Wissenschaftslehre* deduces both how the I sees itself as necessarily determined by its external world and how the I is given freedom of action in this necessarily determined world, i.e., how the I is not *determined* but determining. One should not be deceived here by the rhetoric of freedom: for Fichte, freedom goes hand in hand with necessity. A pluralistic concept of freedom is not part of his philosophy. Human beings are free when they act in accordance with reason, which in turn is not *their individual* reason, but *one* rationality in all, *one* moral law for the freedom *of all*.

### 4. *The Way towards Blessed Life*—Fichte's Berlin Philosophy of Religion (1806)

*The Way Towards Blessed Life* (*Die Anweisung zum seeligen Leben*)—even his contemporaries were surprised by the title and scoffed that Fichte himself must be blessed if he promised to show the way to the blessed life.[3] The enlightened Berlin society did not like the tone in which they were being taught about the blessed life. The audience consisted of government officials, professors, artists, "enlightened Jewish men and women, state councillors, Kotzebue", as Hegel once disparagingly remarked (Hegel 1986, p. 413).

*The Way Towards Blessed Life* has two main parts, the consistency of which is not always clearly recognizable (Cf. Anweisung: Asmuth 1999, 2000; Medicus 1928; Plachte 1922; Schmidig 1966; Seyler 2014; Verweyen 1995; Traub 1992; Seidel 1996). After an introductory and a methodological lecture, Fichte presents the results of his *Wissenschaftslehre* in a first major section—including his doctrine of the *five world-views* (*fünf Weltansichten*). This is followed by an excursus on the Gospel of John. The second part contains Fichte's theory of love, a love with which God loves himself, unfolded in *four stages* of blessedness.

"The very first task of this thinking is: *to think being sharply*." (Fichte et al. 1995, p. 85) Fichte's summary presentation of the *Wissenschaftslehre* begins with the actual and true being *in nuce*. With this opening sentence, Fichte summons the reader to the task. Anyone familiar with Fichte's philosophy knows that the summons (*Aufforderung*) as summons plays a central role in his philosophy. For example, the I can only become a person through reciprocal summoning and being summoned. Here Fichte summons the reader *to think being sharply*. According to Fichte, being should be thought of from two perspectives. *Externally*, it is unchangeable, not becoming, not emerging from something else or previous. It is of itself, from itself, through itself, completely autonomous and unconditioned. *Internally*, being is also unchangeable and eternal, i.e., without reference to time. It does not change and does not become something new. It remains what it was and becomes what it is. Being always remains the same, externally and internally. Being, as Fichte summarizes, must be conceived as one, as "a self-enclosed and perfect and absolutely unchangeable unity" (Fichte et al. 1995, p. 86).

Completely different from this *notion of being* (*Gedanken des Seins*) is the *notion of the existence of being* (*Gedanke des Daseins des Seins*). According to Fichte, *Dasein* or existence means consciousness, representation, revelation, image. Existence is a representation; it is not itself what it is, but has it from another, which it is *not* itself. It refers to something different from itself; it refers to being. Existence is the being of being outside of being. But what does this mean for the notion of being that Fichte summoned us to perform? We have skipped over existence and only believed that "we have come into being itself...; yet we remain, always and forever, only in the forecourt, in existence" (Fichte et al. 1995, p. 87). We have not considered that we think in the thought of being itself, i.e., that we are existence, consciousness. A possible consequence arises here that would be fatal for the concept of *The Way Towards Blessed Life*: there could not be just one single form of the existence of being, but many or even an infinite number of different forms. Being would be different in every existence, and thus also different from itself; for being is absolute *unity*, but existence is *diversity*. Existence is different from every other existence and therefore also different from being. Between being and existence "would rather result in an immeasurable gulf" (Fichte et al. 1995, p. 87). The union between us, existence, and the Absolute, Being or God, would be impossible—impossible for us to attain blessedness.

Existence or consciousness should be the only possible form of existence of being (Cf. Fichte et al. 1995, p. 87). There is a close relationship between being and existence. They appear like two sides of the same coin. Yet they are different, but not in such a way that they completely fall apart. There must not be an unbridgeable gap between them. Fichte formulates the task as follows: Being should be there (*soll da sein*), but it should not lose its absolute character. It should continue to be the Absolute, just as we—under Fichte's guidance—necessarily had to think it. But what kind of thought can manage to sharply separate being and existence on the one hand and make their unity possible on the other? Being, states Fichte, must be "distinguished from existence, and opposed to it; and indeed,—since apart from absolute existence there is nothing else but its existence,—this distinction and this opposition must occur—in existence itself."[4] Thought has made the same mistake as before by skipping over existence and rushing straight to being. We have forgotten ourselves, our thinking and our activity. The distinction between being and existence does not occur somehow, but is only for us and through us. And in distinguishing between them, existence is the active factor. "Existence must grasp, recognize and form *itself* as mere existence (*Das Dasein muß* sich *selber als bloßes Dasein fassen, erkennen und bilden*). The distinction falls in the perspective of existence, which unburdens the absolute being from having to contain a distinction. For absolute being is one, internally and externally without distinction. If one says: existence is distinct from being, then one violates the content of being. Now the distinction does not affect the perspective of being, but that of existence. Existence conceives of itself as existence. What is existence? It is consciousness, imagination, revelation, image, nothing original, but something derived. It exists only in the reference to its original, first, archetypal: being. Just as existence therefore understands

*itself as existence*, it understands itself as the existence of being: it "must, in opposition to itself, posit and form [*setzen, und bilden*] an absolute being, whose mere existence is itself" (Fichte et al. 1995, p. 88). This means that the distinction between being and existence occurs in existence and not—one might add—*in being*. Thus we reach the basic dynamic situation of any *Wissenschaftslehre*: maximum difference with simultaneous maximum pressure for unity, a basic situation that, according to Fichte's program, can only be mediated by Ought (*das Sollen*), an Ought, in turn, that can only be redeemed in the actual act.

Characteristic of Fichte's philosophical language in the years around 1804 is the formulation: existence "must annihilate itself (*sich vernichten*) through *its own* being—in opposition to another absolute existence." This emphatic statement of *self-annihilation* does not mean the radical eradication of consciousness, but is rather intended to indicate that existence or consciousness is nothing independent in itself. Consciousness is *annihilated* insofar as it recognizes itself as consciousness and knows that it means nothing in itself and has no validity.

Existence recognizes itself as existence; existence is consciousness; consciousness recognizes itself as consciousness. It is therefore "self-consciousness of itself (of existence), as a mere image, of the being that exists absolutely in itself."[5] But because it has no meaning in itself, existence cannot jeopardize the unity of being. Existence is the only possible form of being.

Existence conceives of itself. However, it only conceives the factuality of its existence and not how it emerges from the being that is enclosed within itself. For existence cannot comprehend itself beyond itself. It is impossible to ask: What was existence before it was what it is? Fichte states: "[. . .] and so, through the absoluteness of its existence, and through its bondage to this existence, it [=existence] is cut off from all possibility of going above it, and, beyond it, of comprehending and deriving itself...: everywhere it is, it finds itself already there."[6]

Now, if existence cannot emerge from within itself, how does it come to understand itself as the existence of being? Where does existence get the being that is completely independent of and presupposed by it? Outside of being—this is again the beginning of the argument—is nothing and nothing is there. If something is there, then it is there through absolute being. Being and existence are identical in their highest point. "The real life of knowledge is therefore, in its root, the inner being and essence of the Absolute itself, and nothing else; and there is no separation at all between the Absolute, or God, and knowledge, in its deepest root of life, but both merge completely into one another." (Fichte et al. 1995, p. 88)

Fichte's five world-views (*Die fünf Weltansichten*) are: sensuality, lower morality (*Sittlichkeit*), higher morality, religion, and science, i.e., philosophy or: *Wissenschaftslehre* (Cf. Asmuth 1995). These world-views are arranged hierarchically. Fichte emphasizes that each higher form cancels out the validity of its respective lower form. There is thus a clear hierarchy of world views, not a plurality of symbolic forms or versions of the world, as in later considerations of other authors (Cf. Asmuth 2010). Fichte associates lower morality with Kant's practical philosophy and his *Categorical Imperative*, whose function Fichte sees in the fact that the freedom of all can coincide, but lacks positive determinations. The result of low morality is only that I do not have to hold myself in contempt.

Higher morality, by contrast, requires a positive doctrine of ideas, the ideas of the good, the true, and the beautiful, to be pursued for the sake of these ideas themselves. Fichte associates such a doctrine of ideas with the names of Plato and Jacobi: he states that Plato had a notion of it, and Jacobi—one hears the bitter, almost spiteful undertone—sometimes touched it (Cf. Fichte et al. 1995, p. 110). Ultimately, religion consists of recognizing that ideas are grounded in God, that they are nothing more than appearances of God in us, "His expression and His image, totally and absolutely and without any subtraction, thus as His inner being is able to appear in an image." (Fichte et al. 1995, p. 110) But for Fichte, religion does *not* exhaust itself in a mere cognitive position. Instead, religion undermines the discursiveness of theology; indeed, according to Fichte's understanding, religion must

annihilate theology. For the proposition under which all theology, at least that which Fichte knew, must be subsumed, is the proposition—God alone is, and apart from Him there is nothing—this proposition makes God a concept, an empty concept, a meaningless shadow concept, as Fichte insists. But God, Fichte argues, is not a concept at all, but pure life. We ourselves, in so far as we think, are this divine life.

Fichte's view of religion is very similar to various forms of mysticism. And so it is not surprising that some interpretations have devoted a great deal of attention to this similarity (Barion 1929; Ceming 1999. Zur Gegenposition: Cf. Janke 1994; Lasson 1968; Messer 1923). But Fichte's philosophy does *not* end with religion. It does *not* lead to the last wordless word of a path into the absence of language and philosophy in immediate faith and life. In the first part of the *The Way Towards Blessed Life*, Fichte shows that the *Wissenschaftslehre* goes beyond the standpoint of religion, that God is not only manifested in the execution of the moral order of the world, but that He—which is higher—can be clearly seen in pure thought. It is only in philosophy that the genetic connection of all forms of knowledge appears: "Religion, without science, is somewhere a mere faith, even if unshakeable: science abolishes all faith and transforms it into sight."[7]

For Fichte, the standpoint of religion is therefore not an instance at which or towards which one can develop a *philosophy* of religion. Rather, the philosophy of religion is opposed to religion due to its reflective approach. Only the *Wissenschaftslehre* as a universal project embraces the contents peculiar to religion and makes them transparent for knowledge. This is the reason for the ambivalent relationship between the *Wissenschaftslehre* and Christianity. On the one hand, Fichte's writings continue the tradition of the Enlightenment. On the other hand, Fichte strives for a renewal of religion, for a religion of reason. Criticism of religion, criticism of the church, of the clergy, of the tradition and ritual are just as central to Fichte's reflections on Christianity as his efforts to transform Christian beliefs. However, this renewed Christian doctrine is not a reformulated ontology or even metaphysics, but a *Wissenschaftslehre*. This shows the field of conflict in which Fichte's writings on Christianity are situated: the only criterion by which Christianity must be measured is reason. Christianity is true only insofar as it can stand up to philosophy or the *Wissenschaftslehre*. In his diary of 1813 (*Diarium 1813*), Fichte clearly states: "Only the principle of the *Wissenschaftslehre* makes Christianity comprehensible—." (Cf. Meckenstock 1973, p. 67; Danz 2009)

## 5. The Republic of the Germans

One year after *The Way Towards Blessed Life*, i.e., 1807, probably in the spring shortly before his 'escape' from Berlin to Königsberg, Fichte wrote a manuscript: *Science Fiction*: *The Republic of the Germans at the Beginning of the Second and Twentieth Centuries under its Fifth Reich Governor* (*Die Republik der Deutschen zu Anfange des zwei u. zwanzigsten Jahrhunderts unter ihrem fünften Reichvogte*). In the fall of 1806, Prussia had suffered a crushing defeat at the hands of Napoleon's troops (Cf. Asmuth 2021). The old Prussian state collapsed after the double battle of Jena and Auerstedt in October 1806. The court fled to Königsberg. It was only after the decisive defeat against Napoleon at the Battle of Friedland that the war ended for the time being and the Peace Treaty of Tilsit was signed on 9 July 1807, with Prussia losing almost half of its territory. Fichte's strange manuscript, which we know from Fichte's estate, falls within this time frame and provides a good insight into Fichte's thinking between 1805 and 1807. Fichte imagines a historian of the future who looks back on Fichte's present age. He leaves no good hair on the present: "egotistical selfishness", "general wickedness", "Concession", "lack of understanding", "avoidance", "imprudence", "high treason", "heresy", "disgusting flattery": these are the keywords with which the historian of the future looks back. In a kind of report, the historian comments on the reforms that had improved the old bad state of the Germans. These include: "Beauty of public buildings, canals, streets. Churches, schools, avenues. Gardens. This is very important in the cultivation of the country. The Olympic Games. I should be thinking of the designs." (Fichte et al. 1994, p. 389)

"Olympic Games"—Fichte's idea is exceptionally early. In fact, the modern Olympic Games were born in Greece, in the Greek poetry of the publisher Panagiotis Soutsos, who presented the Olympic Games as a symbol of ancient Greek culture in a poem in 1833. Following Greece's independence from the Ottoman Empire (1828) and the intervention of the major European powers, Otto I, a Bavarian prince, ruled Greece as king for 30 years from 1832. Soutsos' efforts were not crowned with success. It took until 1859 for a successful Greek merchant, Evangelos Zappas, to finally organize the first Olympics in Athens with Otto I's permission. Only Greeks were allowed to participate. These Olympics were embedded in a program that offered an industrial and agricultural exhibition, behind which the sports had to take a back seat. The model was the Munich Oktoberfest, also in the middle of the 19th century, an agricultural exhibition accompanied by sports competitions.

Obviously, Fichte's idea differed conceptually from *those* Olympic Games that we have known since 1894. Pierre de Coubertin conceived them as a "meeting of the youth of the world". Fichte was not thinking of a major international sporting event, but was following an antique ideal. His vision follows classicism. The idea is "Olympic Games" of the German states, analogous to the games of the Greek city-states of antiquity. The aim is to form the Germans into a people, into a nation, as Fichte would later propagate in his *Addresses to the German Nation*. He is concerned with nation-building, a unification of the German states and small states under the leadership of Prussia as a counterweight against France and Napoleon. Fichte hoped that German unity would lead not only to a German state with a German constitution, but even more to the unity of the Germans, which he saw as a spiritual unity. Another concern was the military training, a motif that would later become central to Johann Friedrich Ludwig Christoph Jahn, the notorious father of gymnastics, who wanted to revive the *Deutsche Volkstum* through the paramilitary "art of gymnastics". What Fichte conceived entirely out of the depression of Prussia's defeat turned into sheer *völkisch* nationalism. Karl Immermann captured this wonderfully in his *Memorabilien* of 1838/39: "In one of his speeches, Fichte had referred to a closed youth state as a means by which the education of the future generation could become possible. Jahn, who often behaved like Fichte's unconscious monkey, made this fantastic state a reality for a while. It was dominated by an aristocracy of wrestling, swinging, running and gymnastics. The art of gymnastics is a prime example of how a very simple thing can be corrupted and made confusing." (Fichte et al. 2012, p. 265)

The same wind is blowing in Fichte's philosophy of religion, especially that from 1806. Fichte is by no means concerned with an epistemic explanation of acts of faith, nor with a mystical transformation of the Christian religion, nor even with an intellectualistic revision of traditional beliefs (Cf. Asmuth 2022). Instead, Fichte is concerned with an identity-political reformation, if not revolution, of the religious faith of the Germans. The aim is the unity of the German nation in its natural external borders and in its inner consciousness, as he later calls for in his *Addresses*.

In the *Republic of the Germans*, Fichte states that the legislature of the future state had therefore found it necessary—after consulting the scholars, of course—to introduce a fourth confession into the state alongside the Roman-Catholic, Lutheran, and Reformed confessions: the confession of the *General Christians*, and to "elevate them to the proper civic religion, i.e., to the legitimate religion for the task of the state. The *General Christians* recognize the doctrine of God of Christianity, but only insofar as it corresponds to reason (Cf. Die Republik der Deutschen, Fichte et al. 1994, p. 397). As good Fichteans, the *General Christians* consider anything historical about religion to be non-essential. This also applies to Jesus Christ. *General Christians* are not overly interested in the historical person of Jesus. They regard historical controversy as "silliness" (*Albernheit*) (Fichte et al. 1994, p. 397). The center of their religion is reason, the I, "because we ourselves, as the essential, firmly believe to be Christian."[8] This creed is deeply anchored in the constitution of the *Republic of the Germans*, which states, according to Fichte: "The first condition of human education is unrestricted independence, and this consists of recognizing no barriers other than those set by one's own clear insight and firm will. He who must will according to another's

insight is not free. The system of blind faith in authority arises, if not from despots, then certainly from slave minds. But no constitution for a free people may either establish as a constitutional condition of citizenship, or even leave implicit, any decree that appears to set limits to its own insight." (Fichte et al. 1994, p. 414)

Notwithstanding the commonly voiced, Fichte did not derive the idea of an inner national identity from *völkisch* principles like Jahn (Cf. Jahn 1810), nor ultimately from patriotic premises like Ernst Moritz Arndt (Arndt 1806), nor from an anti-Jewish resentment, as Saul Ascher suspected (Cf. Ascher 1794). Instead, the idea stems from a pathos of freedom and reason that is entirely indebted to the style of the French Revolution, which is also fed by transcendental philosophy and thus combines a radicalized and systematically sharpened sense of autonomy in the Kantian sense.[9] In this homogeneous state, there are only *General Christians*. After Fichte's death in 1815, there was a clear break between his view of religion and the identity-political demands of the generation almost fifteen years younger: the Battle of Waterloo led to France's complete military collapse and the Second Peace of Paris in November 1815. The Restoration began, the fraternities and the Germanomaniacs (Cf. Ascher 1815) formed and sharpened the political and nationalist fronts. Karl Immermann can probably be agreed with here, as he sympathetically confessed in his memorabilia: "If anyone died at the right hour for his rest, it was Fichte." (Fichte et al. 2012, p. 265)

**Funding:** This research received no external funding.

**Data Availability Statement:** Not applicable.

**Conflicts of Interest:** The author declares no conflict of interest.

## Notes

[1] "The spirit [*Geist*] requires a general conception [*Vorstellung*] of the purpose [*Zweck*], of the determination of the whole, in order to know what to expect. One wants to have a general view of the landscape, which one then loses sight of when one begins the journey into the various parts" (Hegel 1986, p. 25, Anm. 10).

[2] Cf. for the justification of this view: (Asmuth 2018).

[3] This was reported by Fichte's wife, Marie Johanne. See: (Fichte et al. 1981, p. 276).

[4] (Fichte et al. 1995, p. 87f): "Es muß von dem Daseyn, unterschieden, und demselben entgegengesetzt werden; und zwar,—da außer dem absoluten Seyn schlechthin nichts anderes ist, als sein Daseyn,—diese Unterscheidung, und diese Entgegensetzung muß—In dem Daseyn selber—vorkommen."

[5] (Fichte et al. 1995, p. 88): "Selbstbewußtseyn seiner (des Daseyns) selbst, als bloßen Bildes, von dem absolut in sich selber seyenden Seyn".

[6] (Fichte et al. 1995, p. 88): "und so ist ihm denn durch die Absolutheit seines Daseyns, und durch die Gebundenheit an dieses sein Daseyn, alle Möglichkeit über dasselbe hinauszugehen, und, jenseit desselben, sich noch zu begreifen, und abzuleiten, abgeschnitten . . .: allenthalben wo es ist, findet es sich schon vor".

[7] (Fichte et al. 1995, p. 112): "Die Religion, ohne Wissenschaft, ist irgendwo ein bloßer, demohngeachtet jedoch, unerschütterlicher, Glaube: die Wissenschaft hebt allen Glauben auf, und verwandelt ihn in Schauen".

[8] (Fichte et al. 1994, p. 398): "weil wir selbst, als dem Wesentlichen, Christuße zu sein fest glauben".

[9] On the systematic significance of transcendental philosophy cf.: (Asmuth 2023).

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
