# Peer review of "Revolution and Nation: Johann Gottlieb Fichte’s Late Philosophy of Religion"

_religions, doi:10.3390/rel15040426_

Round 1

Reviewer 1 Report

Comments and Suggestions for Authors

Please see file. Thank you!

Author Response

I have explained the thesis once again in a new paragraph. Furthermore, I have made the references to the authors Asmuth and Traub requested by the reviewer. The detailed interpretations and detailed analyses requested by the reviewer can be found in the places indicated and do not need to be repeated.

Reviewer 2 Report

Comments and Suggestions for Authors

This is an interesting article on a crucial topic in the formation of German idealism. I recommend its publication. However, the text can still be improved. For example, when on page 3 it is stated that "Fichte´s conception in Jena and later in Berlin is very similar to the religious cult of the French Revolution, the cult of reason", it seems to me that it is not sufficiently argued. Moreover, the Fichtean conception should rather be connected with a whole process of de-dogmatisation of the Christian religion, in which the young idealists and Romantics of the time, as well as Kant himself, participate, and which has more to do with pietism-type religious movements than with the revolutionary ideal of a religion of reason. For the rest, a deeper dialogue with up-to-date secondary literature is lacking. 

Author Response

I have explained the thesis again in a paragraph at the beginning of the essay. Furthermore, I have used a few references to current authors (Asmuth, Traub, Danz) to clarify the integration of my thoughts into the current state of the discussion on the interpretation of Fichte's philosophy of religion.

Reviewer 3 Report

Comments and Suggestions for Authors

This is what I say to the editors. In short, I think this is a good and concise article, valuable for the reflection it inspires in the reader. However, the author should at least 1) correct the statement that the 'disciplining' of philosophy is extrinsic to Fichte's aims, 2) correct the view that Kant's philosophy admitted no ontological substrate, and 3) give more attention to the explicit statement of a thesis and organization of the article's discussion with respect to that thesis. There are more detailed comments in my letter; I hope the author attends to them. I hope this can improve the article, because it is good enough to deserve this improvement.

Comments on the Quality of English Language

There is good English usage overall, although I felt line 38 to be confusing and no doubt other lines feature sentences that could be improved.

Author Response

(The authors gave the same response as above.)
